# Observed and CMIP6 model simulated organic aerosol response to drought in the contiguous United States during summertime

Wei Li[1,2] and Yuxuan Wang[1]

[1]Department of Earth and Atmospheric Sciences, University of Houston, Houston, Texas, USA

[2]Now at Cooperative Institute for Satellite Earth System Studies, George Mason University, Fairfax, Virginia, USA

*Corresponding author: Yuxuan Wang (ywang246@central.uh.edu)*

**Abstract.** Drought events have been linked with the enhancements of organic aerosols (OA), but the mechanisms have not been comprehensively understood. This study investigates the relationships between the monthly standardized precipitation–evapotranspiration index (SPEI) and surface OA in the contiguous United States (CONUS) during the summertime from 1998 to 2018. OA under severe drought conditions shows a significant increase in mass concentrations across most of the CONUS relative to non-drought periods with the Pacific Northwest (PNW) and Southeastern United States (SEUS) experiencing the highest average enhancement of $1.79\ \mu g\ m^{-3}$ (112 %) and $0.92\ \mu g\ m^{-3}$ (33 %), respectively. In the SEUS, a linear regression approach between OA and sulfate was used to estimate the epoxydiols-derived secondary organic aerosol (IEPOX SOA), which is the primary driver of the OA enhancements under droughts due to the simultaneous increase in biogenic volatile organic compounds (VOCs; such as isoprene and monoterpene) emissions and sulfate. The rise of sulfate is mainly caused by the reduced wet deposition because of the up to 62% lower precipitation amount. In the PNW, OA enhancements are closely linked to intensified wildfire emissions, which raise OA mass concentrations to be four to eight times higher relative to non-fire conditions. All ten Earth system models participating in the sixth phase of the Coupled Model Intercomparison Project (CMIP6) can capture the slopes between SPEI and OA in the PNW with CESM2-WACCM and GFDL-ESM4 performing the best and worst in predicting the OA enhancement under severe droughts. However, all models significantly underestimate the OA increase in the SEUS with Nor-ESM2-LM and MIRCO6 showing relatively better performance. This study reveals the key drivers of the elevated OA levels under droughts in the CONUS and underscores the deficiencies of current climate models in their predictive capacity for assessing the impact of future droughts on air quality.

## 1. Introduction

Drought events, marked by prolonged periods of water scarcity and precipitation deficits, have profound impacts on the hydrological cycle, ecosystems, and society (Wilhite et al., 2007). The contiguous United States (CONUS) is especially prone to droughts, and recent years have witnessed an escalation in both the frequency and severity of drought episodes across various regions (Leeper et al., 2022; Strzepek et al., 2010). These drought events are intricately linked to the modifications in atmospheric processes, such as emission, production, transport, and deposition, which can extend beyond the immediate hydrological impacts with far-reaching implications for air

quality. Specifically, organic aerosol (OA), a major component of the particulate matter with an aerodynamic

diameter less than or equal to 2.5 μm ($PM_{2.5}$), emerges as a critical air quality concern influenced by the complex

interactions between drought-induced meteorological conditions and biogeochemical processes.

OA can be directly emitted into the atmosphere through combustion activities, such as transportation fuel and

biomass burning. This kind of OA is called primary organic aerosol (POA), whereas secondary organic aerosol

(SOA) is produced by the oxidation of volatile organic compounds (VOCs). The intricate interplay between drought

and OA dynamics involves complex feedback mechanisms. Biogenic isoprene, mainly emitted by terrestrial

vegetation, is an important precursor of SOA and is highly sensitive to drought conditions. Both laboratory and field

measurements have shown that biogenic emissions of isoprene will increase at the initial stage of drought

development primarily due to temperature stimulus but drop eventually under prolonged severe drought limited by

soil water availability (Pegoraro et al., 2005; Brilli et al., 2007; Potosnak et al., 2014). The abnormally high

temperature and low humidity under droughts can enhance the oxidation of OA (Maria et al., 2004; Yli-Juuti et al.,

2021), while low cloud water content lowers the aqueous SOA formation (Brégonzio-Rozier et al., 2016; Tsui et al.,

2019), leading to compensating changes in the mass and hygroscopicity of OA. Aerosols are most effectively

removed by wet scavenging, which will be reduced under lower rainfall intensity and frequency (Dawson et al.,

2007; Fang et al., 2011). In addition, dry conditions can trigger large and high-intensity wildfires, emitting more

POA and VOC precursors into the atmosphere (Ruffault et al., 2018; Taufik et al., 2017). The interactions of these

factors underscore the need for a comprehensive understanding of the mechanisms driving variations in OA during

drought events.

OA, due to its fine particulate nature and diverse chemical composition, exerts significant adverse effects on climate

and human health. OA is found to be associated with a higher county-level cardiorespiratory mortality rate than

other major $PM_{2.5}$ components, such as sulfate, ammonium, and nitrate (Pye et al., 2021). OA can scatter solar

radiation, form cloud condensation nuclei, and affect cloud droplet concentrations, posing big uncertainties on

radiative forcing and climate feedback (Carslaw et al., 2013; Lee et al., 2016). The coupled chemistry-climate

models and Earth system models (ESMs) are fundamental tools for studying global warming and the accuracy of

OA simulations in these models are crucial constraints on their credibility in climate change simulation and

projection (Gomez et al., 2023; Thornhill et al., 2021). The Coupled Model Intercomparison Project Phase 6

(CMIP6), containing the new generation of ESMs with interactive aerosol and gas chemistry implemented (Turnock

et al., 2020), provides a valuable opportunity to evaluate the simulated OA and its response to drought, which is

projected to be more frequent in the future (Cook et al., 2018)

Several case studies have focused on the impacts of droughts on the concentrations and speciation of $PM_{2.5}$ in the

CONUS by calculating the differences between drought and non-drought years (Wang et al., 2015; Borlina and

Rennó, 2017; Zhao et al., 2019). Wang et al. (2015) and Zhao et al. (2019) compared the concentrations of $PM_{2.5}$

and its compositions in the southern/southeastern U.S. during the severe drought in the 2011 summertime against the

non-drought year of 2010 and 2013, respectively. They show that $PM_{2.5}$ has a respective enhancement of 47% and

65% with the largest contribution from the increase of organic carbon (OC) by 119% and 117%. Following OC,
sulfate in the southeast US is enhanced by 84% during the 2011 drought relative to 2013. However, fewer studies
have carried out long-term analyses, which can help derive a more robust drought-aerosol association than case
studies. Wang et al. (2017) performed a 25-year analysis during the growing season (March-October) from 1990 to
2014 and found that, on a monthly scale, the overall 17% enhancement of $PM_{2.5}$ in the CONUS is mainly attributed
to the increase of OA, sulfate, and dust. Each of these species has a unique spatial pattern in their response to
droughts, which warrants a further subregional analysis to reveal the processes causing such spatial distribution
discrepancy.
In this study, we focus on the changes in OA under droughts over the CONUS during the study period of
summertime from 1998 to 2018. Spatial patterns of the responses of OA to droughts will be explored, followed by a
regional analysis focusing on the southeastern US (SEUS) and Pacific Northwest (PNW) where the highest
responsive rates of OA to droughts are found. The processes responsible for the increase of OA in these regions will
be discussed. At last, the observed drought-OA relationships will be used as a process-level metric to evaluate OA
simulations in the CMIP6 ESMs, which can shed light on future model development and improve aerosol
predictions.
**2. Datasets**
**2.1 Drought indicator**
The one-month gridded Standardized Precipitation-Evapotranspiration Index (SPEI) data from the global SPEI
database (https://spei.csic.es/, last access: November 27, 2023) was selected as the drought indicator because of its
numerical nature allowing for statistical analysis (e.g., correlation and regression). The SPEI is a multi-scaler index,
allowing for the identification and comparison of drought severity through time and space (Vicente-Serrano et al.,
2010). Negative values of SPEI are indicative of droughts and vice versa. The dataset has a spatial resolution of 0.5°
× 0.5° and a temporal range of 1973-2018. A composite analysis can also be conducted by applying the criteria of
SPEI < -1.3 and SPEI > -0.5 to denote severe drought and non-drought conditions, respectively, as suggested by Wang
et al. (2017).
**2.2 Air quality and meteorological data**
To expand the spatial coverage, we created a gridded daily organic carbon (OC) dataset (0.5° × 0.5°) from 1998 to
2018 that aggregates site-based observations from the Interagency Monitoring of Protected Visual Environments
(IMPROVE) network using the modified inverse distance weighting method as done by Schnell et al. (2014). Data
from the IMPROVE sites has been widely used by previous studies to investigate surface particulate matter trends or
variations in the CONUS (e.g., Hand et al., 2012). A factor of 2.1 was used to convert OC observations to OA as
suggested by other studies (Pye et al., 2017; Schroder et al., 2018). US Environmental Protection Agency Chemical
Speciation Network (EPA-CSN) also provides long-term OA data, but the CSN network uses different sampling
practices and analytical methods from IMPROVE, which can lead to systematic differences in OA measurements
(Hand et al., 2012; Gorham et al., 2021). Thus, we only used the IMPROVE dataset in this study. To reduce the artifact
caused by different data completeness (e.g., old sites retired and new sites started), we selected the sites with data
records longer than 5 years during the study period for interpolation following Li and Wang (2022). Based on this
criterion, there are a total of 175 sites selected for interpolation, ~80% of which have a data record equal to or greater
than 15 years, suggesting small temporal uncertainties caused by the spatial interpolation (Figure S1).
Sulfate is known to influence the formation of epoxydiols derived secondary organic aerosol (IEPOX SOA), a key
component of OA. To explore how this linkage changes with drought, we generated a gridded sulfate dataset following
the same method as OC. Monthly sulfate wet depositions with associated precipitation amount and pH were obtained
from the National Atmospheric Deposition Program (NADP). There is a total of 53 NADP sites in the SEUS (defined
in Section 3.1) with a more than 5-year data record during the study period. We obtained the satellite-based low level
(below 700 hPa) cloud cover and liquid water content (LWC) between 2000 to 2018 from the Clouds and the Earth's
Radiant Energy System (CERES) monthly Single Scanner Footprint $1° \times 1°$ (SSF1deg) product
(https://asdc.larc.nasa.gov/project/CERES/CER_SSF1deg-Month_Terra-MODIS_Edition4A, last access: November
28, 2023). To investigate OA changes from wildfire, monthly open fire emissions were from the Global Fire Emission
Database version 4 (GFED4) for 1998–2018 (Giglio et al., 2013). The version of GFED4 we used includes the burned
area contributions from small fires, which increases the total amount of burned area by 75% relative to its previous
version and brings the prescribed burned area estimates into closer agreement with those reported by the National
Interagency Fire Center (Randerson et al., 2012). Thus, the prescribed fire burning is partly, if not all, considered in
the analysis.
**2.3 CMIP6 AerChemMIP models**
Ten models from the CMIP6 Aerosol Chemistry Model Intercomparison Project (AerChemMIP) were selected: BCC-
ESM1, CESM2-WACCM, CNRM-ESM2-1, EC-Earth3-AerChem, GFDL-ESM4, GISS-E2-1-G, MIROC6, MRI-
ESM2-0, NorESM2-LM, and UKESM1-0-LL. They are the only models found by the time of writing with OA and
sulfate mass concentration outputs from historical simulations with prescribed sea surface temperature in the
AerChemMIP project from 1850 to 2014. No ensemble members were found for the ten models. Various aerosol
schemes are used by the models with different treatments for gas phase reactions and secondary aerosol formation.
More information and references (Danabasoglu et al., 2020; Dunne et al., 2020; Kelley et al., 2020; van Noije et al.,
2021; Séférian et al., 2019; Seland et al., 2020; Senior et al., 2020; Tatebe et al., 2019; Wu et al., 2020; Yukimoto et
al., 2019) for each model are listed in Table S1.
**3. Results**
**3.1 Spatial Distributions of Organic Aerosol Response to Drought**
Figure 1a shows the maps of the mean summertime (JJA 1998–2018) surface OA concentrations under non-drought
conditions and their changes under severe droughts with the observational sites (dots) overlaid. The associated
frequency and OA standard deviation during non-drought and severe drought periods are displayed in Figure S2.
The western US states along the Rocky Mountains exhibit the highest severe drought frequency of up to 25%, while
wet and normal conditions are more common in the eastern US and southern California with a frequency of more
than 80%. Higher OA concentrations can be found in central California and the eastern US under non-drought
conditions, reflecting the average spatial distributions of summertime OA. Under severe droughts, most of the grids
and sites display an enhanced OA level with a mean increase of 0.72 $\mu g\,m^{-3}$ across all the grids and 0.78 $\mu g\,m^{-3}$
across all the sites in the CONUS. Higher enhancements occur in the Pacific Northwest (PNW; 42-50°N, 105-
125°W; red box in Figure 1a) and southeast U.S. (SEUS; 25-37°N, 75-100°W; blue box in Figure 1a). In both
regions, the overall gridded OA statistical distributions under severe droughts move towards the higher end
compared with those under non-drought conditions (Figure 1b), with an increase in the mean value by 1.79 $\mu g\,m^{-3}$
(112 %) and 0.92 $\mu g\,m^{-3}$ (33 %) across the PNW and SEUS, respectively. Similar results are found using on-site
data with a respective increase of mean value by 2.18 $\mu g\,m^{-3}$ (118 %) and 1.11 $\mu g\,m^{-3}$ (34 %), which indicates the
interpolation does not significantly affect the results. OA experienced a downward trend in the SEUS during the last
two decades due to the reduction of anthropogenic emissions (Ridley et al., 2018). To verify whether the trend will
significantly affect our results in the SEUS, we reproduced Figure 1b in Figure S3a using detrended OA. The
detrend is conducted by removing the 7-year moving average from the raw data in the same month of each year
following Wang et al. (2017) and Li et al. (2022). OA enhancement under severe droughts is 0.78 $\mu g\,m^{-3}$ and 1.02
$\mu g\,m^{-3}$ for gridded and on-site data, respectively, which is comparable to those values derived from raw OA data in
the SEUS area. This indicates that anthropogenic emission changes do not significantly interfere with our analysis
and instead natural processes play a more important role in causing the enhancement of OA in the SEUS region.

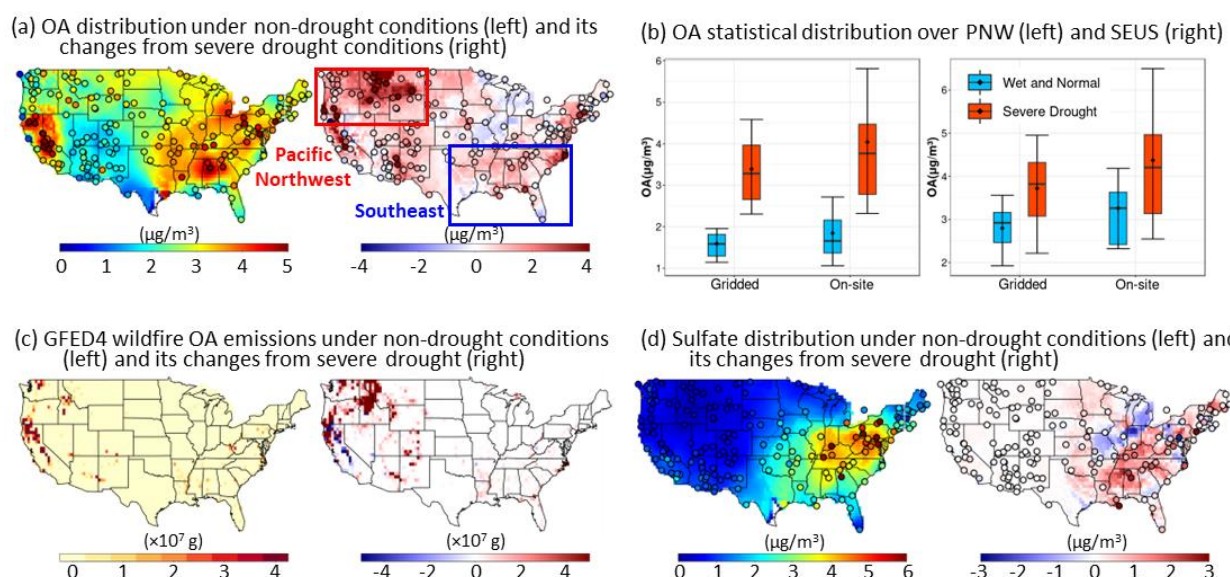

**Figure 1. (a) Maps of the mean gridded and in situ (dots) OA under non-drought (wet and normal) conditions (left) from**
**1998 to 2018 in JJA and its changes from severe drought conditions (right). (b) Comparisons of statistical distributions of**
**gridded and on-site OA mass concentrations under severe drought (red boxes) and non-drought (blue boxes) conditions**
**over the Pacific Northwest (left) and southeast region (right). (c-d) Same as a, but for OA monthly wildfire emissions from**
**GFED4 inventory and sulfate, respectively.**
Wildfire, a major source of biomass burning, is one of the biggest contributors to both POA and SOA globally
(Hallquist et al., 2009; Gilman et al., 2015; Jen et al., 2019). In the western U.S., OA, as the largest component of
$PM_{2.5}$, experiences an upward trend, opposite to the rest of the country, due to the increasingly higher wildfire
frequency (Dennison et al., 2014; McClure & Jaffe, 2018; Wang, et al., 2022). Indeed, we found many 'hot spots' of
wildfire emissions of OA over the western U.S. under non-drought conditions based on the GFED4 wildfire fire
inventory (Figure 1c). Severe droughts can lead to extremely high wildfire OA emissions over the PNW region,
which corresponds to the highest OA enhancement and variability as shown in Figure 1a and Figure S2b,
respectively. In contrast, the SEUS undergoes a much lower enhancement of wildfire OA emissions under severe
droughts. Biogenic secondary organic aerosol (BSOA) is reported to be the major fine aerosol component in the
SEUS, accounting for 60%–90% of the total $PM_{2.5}$, due to the abundant isoprene emissions (Zhang et al., 2012;
Hidy et al., 2014; Kim et al., 2015). The concentrations of BSOA in the SEUS region strongly depend on ambient
sulfate through the reactive update of gas-phase epoxydiols (IEPOX) onto the aqueous acidified surface of sulfate
particles (Surratt et al., 2010; Xu et al., 2015; Lopez-Hilfiker et al., 2016; Malm et al., 2017). Interestingly, the
highest sulfate increase during drought is found in the SEUS (Figure 1d), presumably due to enhanced gas-phase
sulfate production and reduced wet deposition (Wang et al., 2015; Xie et al., 2019). The higher sulfate
concentrations during droughts lead to the enhanced formation of IEPOX SOA, which is likely an important factor
leading to a higher OA level in the SEUS.

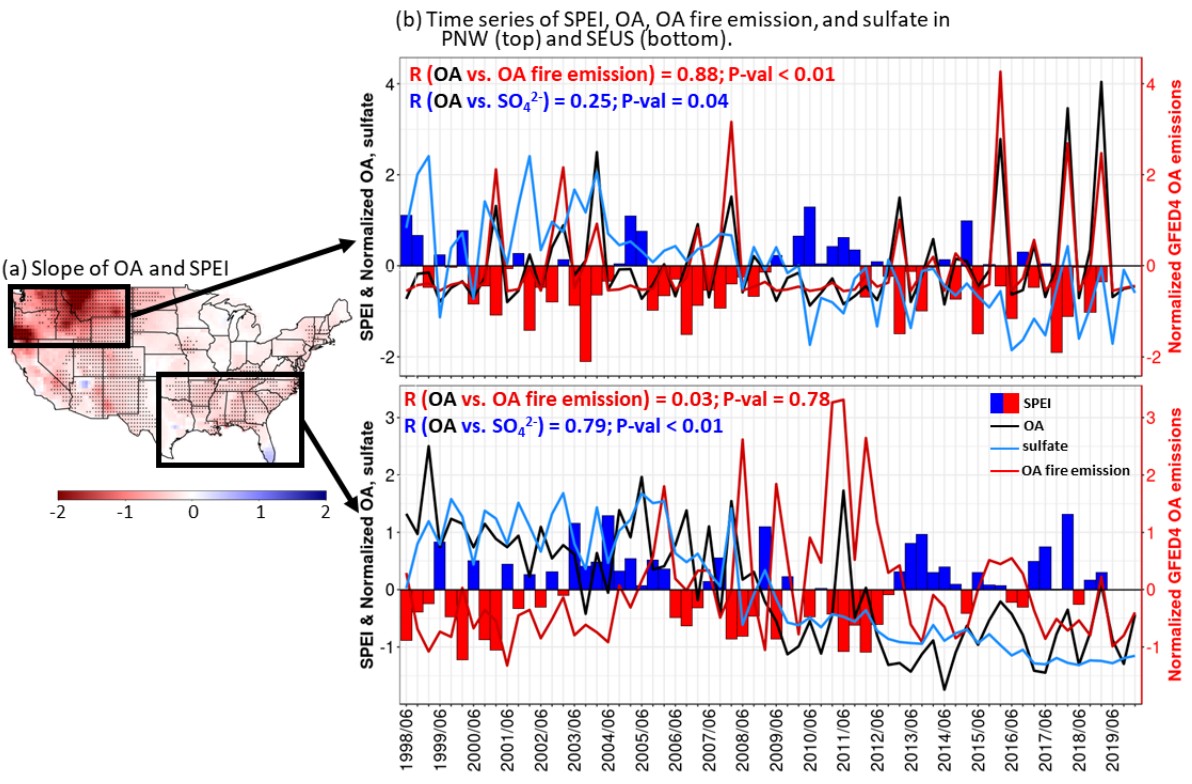


**Figure 2. (a) Map of the slopes between monthly gridded OA and SPEI. Black dots indicate the slopes with P-vales less than**
**0.05. (b) Time series of SPEI (bar), normalized OA (black line), sulfate (blue line), and wildfire OA emissions from GFED4**
**inventory (red line; right axis) averaged across the PNW (top) and SEUS (bottom) region. The numbers indicate the**
**correlation coefficient (R) and P-value (P-val) between OA and sulfate (blue) and wildfire emissions (red).**

Using the numerical drought indicator of SPEI, we calculated the linear slopes between monthly OA and SPEI in
each grid (Figure 2a). Consistent with the composite analysis in Figure 1a, most of the grids show negative slopes
with the highest absolute values of more than $2 \ \mu g \, m^{-3}$ per unit change of SPEI occurring in the PNW region. It is
noteworthy that negative values of SPEI indicate droughts, and thus the negative slopes with SPEI signify an
enhanced OA level over most of the CONUS during drought. We further examined the monthly time series of the
regional mean of SPEI, normalized OA, sulfate, and OA wildfire emissions in the PNW and SEUS (Figure 2b). OA
in the PNW region is strongly correlated with OA emissions from fire with a high correlation coefficient (R) of 0.88.
The extremely high values of OA and OA fire emissions are also concurrent with droughts when SPEI is negative
(red bars). On the contrary, SEUS has a weak correlation between OA and OA fire emissions yet a high association
between OA and sulfate with an R value of 0.79. Wildfire seems only to have high contributions to peak OA values
in extreme drought years, such as in 2011. Based on the correlation coefficients, more than 60% and 70% of the
monthly OA variability can be explained by sulfate and wildfire emissions in the SEUS and PNW regions,
respectively, which deserves an in-depth exploration in the next section.

**3.2 Regional Analysis in the Pacific Northwest and Southeast US**
In this section, we conducted a regional analysis of OA, focusing on OA relationships with sulfate in the SEUS and
with wildfire emissions in the PNW. In the SEUS, we calculated the linear regression between OA and sulfate in
Figure 3a following the method of Malm et al. (2017). Each data point represents the SPEI bin-averaged value of
OA and sulfate from each grid cell. The bins are divided to have approximately the same number of samples
following Xie et al (2019). Only the grids with all five SPEI bins present are used (N=673), which include more than
95% of the total grids (687). Thus, the binned regression calculation can represent the regional conditions of each
SPEI bin. The resulting linear lines and formula are also displayed in Figure 3a. Here the slope calculation is
different from Zheng et al. (2020), in which they averaged OA and sulfate across all the sites in the SEUS and
performed the linear regression temporally. We adopted a spatial calculation of the linear slopes for two reasons: (1)
Averaging across all the sites/grids will significantly reduce the number of data points after the allocation among
SPEI bins; (2) The regional mean of SPEI may average out some drought signals because drought is grid specific
and can differ spatially within the SEUS (Ford et al., 2014). Despite the different methods used, the linear slope in
our calculation (0.56) under non-drought conditions is similar to that of Zheng et al. (2020) using SEARCH
(SouthEastern Aerosol Research and Characterization) sites (0.51). Therefore, our linear slope calculation method
reproduces the sensitivity of OA to sulfate reported by the existing studies.
As SPEI changes from positive (non-drought) to negative (drought), the slope between OA and sulfate becomes
increasingly higher, ranging from 0.56 to 0.79. This indicates more OA formations per unit increase in sulfate as
drought severity intensifies. Although high correlations do not necessarily indicate causal relationships, the chemical
mechanism of IEPOX SOA formation with the presence of sulfate is well documented (e.g., Shrivastava et al.,
2017). The higher sensitivities of OA to sulfate under droughts can be explained by the increasingly higher isoprene
concentrations as shown in our previous studies in the SEUS (Li et al., 2022; Wang et al., 2022b), resulting in more
IEPOX in the atmosphere to be further converted to particle phase catalyzed by sulfate. In addition, the formation of
monoterpene-derived organosulfates, a major component of IEPOX SOA, is also dependent on sulfate (D'Ambro et
al., 2019) and the biogenic emissions of monoterpenes are likely to be intensified during droughts (Llusià et al.,
2008; Wu et al., 2015). Organosulfates originated from anthropogenic precursors are also reported by some studies
(Riva et al., 2015; Le Breton et al., 2018), but they are mainly found in highly polluted urban areas. We further
reproduced Figure 3a using detrended OA and sulfate data, which can remove the effects of anthropogenic
emissions (Figure S3b). A similar pattern of the gradually increasing slope from the wettest (slope=0.18) to the
driest (slope=0.48) SPEI bin was found, which verifies the stronger dependence of OA on sulfate under droughts is
mainly caused by biogenic sources.

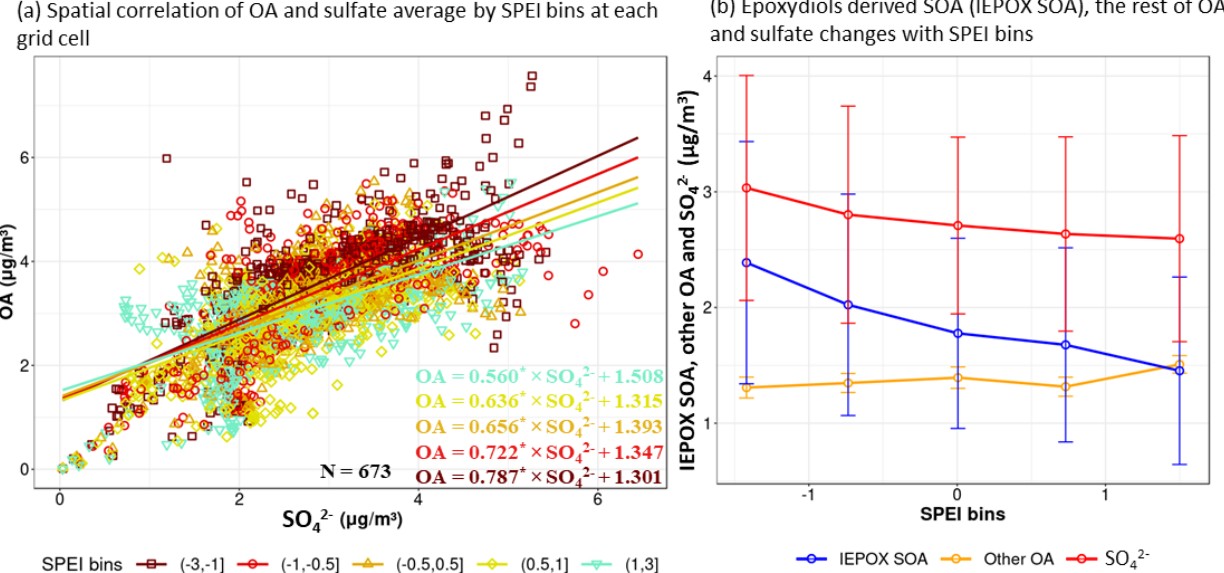


**Figure 3. (a) Scatter plot of the SPEI bin-averaged sulfate and OA at each grid in the SEUS with solid lines representing the linear regressions of OA and sulfate. The corresponding linear formula of each SPEI bin is listed in the bottom-right corner with N indicating the number of data points for each regression calculation. The star marks in the formula indicate the regression significance at a 95% confidence level. (b) The epoxydiols derived SOA (IEPOX SOA), other SOA, and sulfate changes with SPEI derived from the linear regressions in a. Vertical bars indicate one standard deviation.**

The intercept of the linear regression can be interpreted as other OA components that are not associated with sulfate-
catalyzed IEPOX SOA, such as POA and anthropogenic SOA (Malm et al., 2017). Figure 3b shows that the
intercepts (other OA) are stable among the five SPEI bins with a less than 0.2 µg m$^{-3}$ (15%) difference. The
differences of regional mean OA minus the intercepts can then be considered as IEPOX SOA related to sulfate. The
resulting estimate of IEPOX SOA is 1.45 µg m$^{-3}$, 1.68 µg m$^{-3}$, 1.78 µg m$^{-3}$, 2.02 µg m$^{-3}$ and 2.39 µg m$^{-3}$ for the five
SPEI bins ranging from wet to dry conditions. These values correspond to an increase of 0.30 µg m$^{-3}$ IEPOX SOA
per unit decrease in SPEI. Interestingly, there is also an increasingly higher sulfate level from wet to dry SPEI bins
with a mean value of 2.59 µg m$^{-3}$, 2.63 µg m$^{-3}$, 2.71 µg m$^{-3}$, 2.80 µg m$^{-3}$ and 3.03 µg m$^{-3}$, respectively,
corresponding to an overall increase rate of 0.14 µg m$^{-3}$ sulfate per unit decrease of SPEI. Therefore, the increase of
OA in the SEUS under droughts is largely caused by the boosted formation of BSOA due to the concurrent increase
in VOC emissions and sulfate. This is consistent with the modeling case study by Zhao et al. (2019) who found that
98% of the SOA increase during drought in the SEUS is of biogenic origin. It is noted that the approximation of
IEPOX SOA here is the upper limit of BSOA since other processes that can lead to the simultaneous changes of
sulfate and OA, such as wildfire, are miscounted as BSOA in the calculation. Further analysis is needed to attribute
the changes of SOA to different sources more accurately.

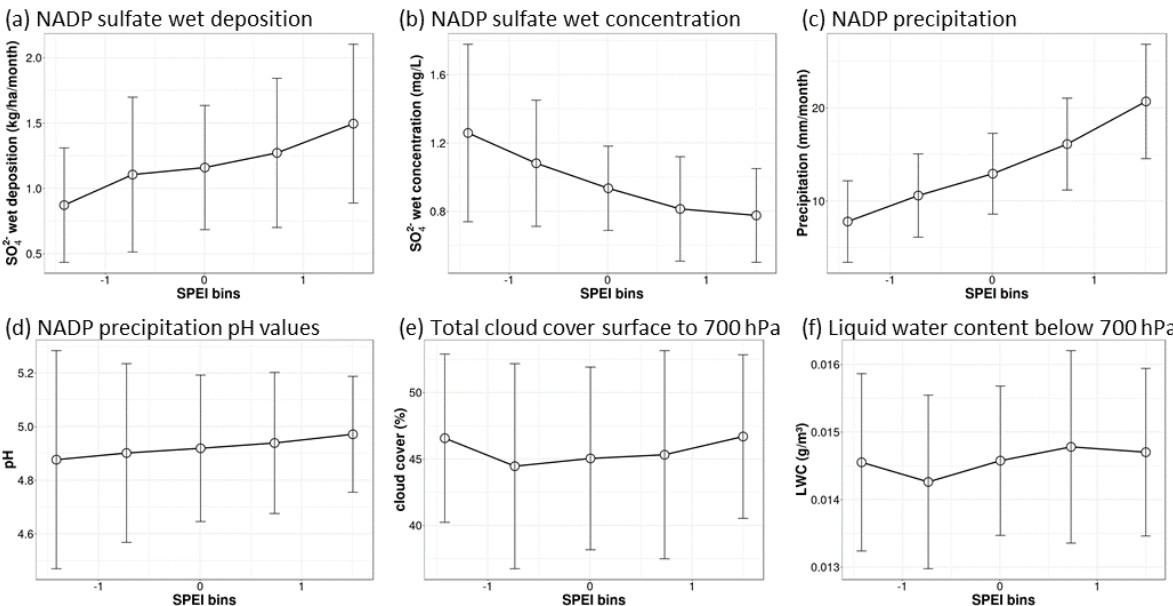

**Figure 4. SPEI bin-averaged sulfate wet deposition (a), wet concentration (b), precipitation amount (c), precipitation pH**
**values (d) from the NADP network, and the total cloud cover (e) and liquid water content (LWC; f) below 700 hPa from**
**the MODIS satellite in the SEUS. Vertical bars indicate one standard deviation.**
The source and sink of atmospheric sulfate are greatly affected by clouds and precipitation because most of the
sulfate is formed in clouds and efficiently removed by wet scavenging (Barth et al., 2000; Rasch et al., 2000; Berg et
al., 2015). Thus, it is understandable that sulfate is sensitive to drought considering both clouds and precipitation are
significantly modulated under droughts. To further investigate the processes causing the increase of sulfate, we
analyzed sulfate wet deposition, wet concentration, precipitation amount, and pH values (Figure 4a-d) from the
NADP network. There is a decreasing trend of sulfate wet deposition from 1.50 kg ha$^{-1}$ month$^{-1}$ at the wettest (SPEI
> 1) to 0.87 kg ha$^{-1}$ month$^{-1}$ at the driest (SPEI < -1) level. The corresponding reduction in precipitation is 62%.
Since sulfate wet deposition is calculated using sulfate wet concentration weighted by precipitation, the 50%
decrease of sulfate wet deposition is driven by the reduced precipitation, which outweighs the increase of sulfate
concentrations.
The low level (below 700 hPa) cloud cover and liquid water content (LWC) are not highly sensitive to droughts with
less than 2% and 4% changes among the five SPEI bins, respectively (Figure 4e-f). Thus, the increase of sulfate wet
concentrations in precipitation is likely indicative of an enhanced formation of aqueous sulfate in the clouds, which
then precipitates. Alternatively, gas phase production of sulfate can also be elevated under droughts due to more
sulfur dioxide ($SO_2$) emissions (e.g. from increased electricity generation and fires) and higher temperatures (Tai et
al., 2010; Wang et al., 2017), and then washed out by rainwater droplets causing higher sulfate wet concentrations in
precipitation. Either of these two pathways suggests that there is higher sulfate formation under droughts which
contributes to the enhanced sulfate besides reduced wet deposition. Furthermore, the mean pH value drops steadily
with dryness levels from 4.98 to 4.87, which further intensifies the acid-catalyzed IEPOX ring opening and leads to
faster BSOA formation (Surratt et al., 2010). Although the rate of IEPOX SOA formation is slower in cloud water
compared to aerosol particles due to its relatively higher pH values (Gaston et al., 2014), the large liquid water
content of clouds, which promotes dissolution, could lead to significant IEPOX SOA formation. Based on a box
model simulation conducted by Tsui et al (2019), increasing pH values in cloud water while keeping the other
factors constant results in a slower rate of IEPOX SOA formation. Additionally, cloud water processing at pH $\leq$ 4
can produce more IEPOX SOA than aerosol particles. Despite the average pH value of ~5 across the SEUS region,
some sites may experience more acidic rainwater in drought months. During the study period, we found two sites in
Georgia and North Carolina with pH less than 4 and their corresponding SPEI values are –0.98 and –1.39.
Therefore, droughts are likely to reduce cloud pH values lower enough at some locations and favorable for
significant IEPOX SOA formation.
Using the same approach as in the SEUS, we calculated the SPEI bin-averaged OA and OA wildfire emissions from
the GFED4 inventory in the PNW region shown in Figure 5. OA fire emissions grow from $0.09 \times 10^7$ g per month at
the wettest level to $4.94 \times 10^7$ g per month at the second driest level (SPEI between -1.5 and -1), followed by a small
drop to $4.17 \times 10^7$ g per month at the driest level (SPEI less than -1.5). This drop is likely caused by the reduction in
the supply of fire fuel load under extreme drought conditions (Scasta et al., 2016). Overall, OA fire emissions
increase by $1.44 \times 10^7$ g per unit decrease of SPEI per month. The mass concentrations of OA resemble the changes
of OA fire emissions with an overall increase rate of 1.01 $\mu$g m$^{-3}$ per unit decrease of SPEI, which indicates more
wildfire emissions are the major driver of the higher OA concentrations in the PNW.

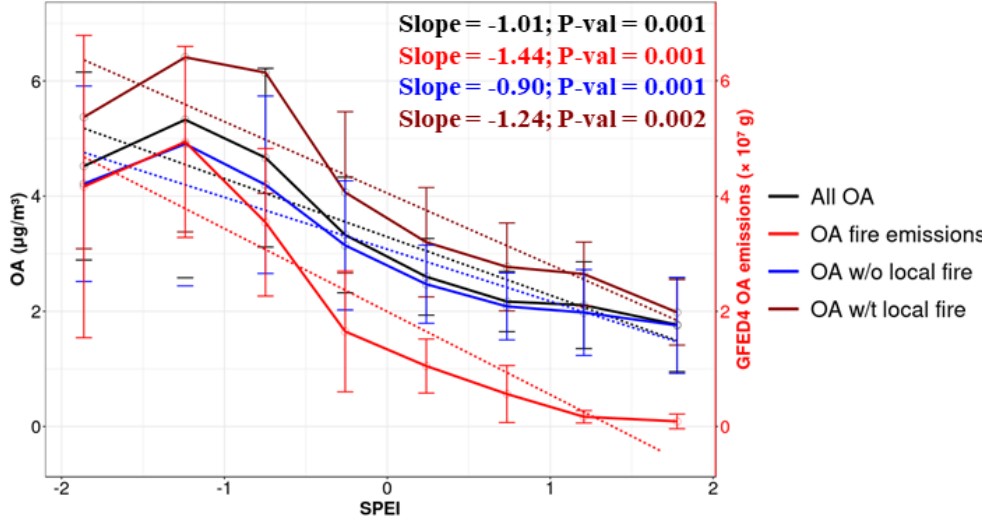

**Figure 5. Mean (point) and one standard deviation (vertical bar) of OA (black line), wildfire OA emissions from GFED4**
**inventory (bright red line; right axis), and OA with (dark red line) and without (blue line) local fire occurrence within each**
**SPEI bin. The dashed lines represent the linear regression with the slopes (Slope) and P-values (P-val) of each variable**
**listed in the top-right corner.**
To better quantify the contributions of wildfire, we further separated OA values into those with local fire influences
if OA fire emissions are greater than zero at each grid in each month and those without local fire influences if zero
fire emissions are found. The time series of OA grouped by periods with and without wildfire emissions within each
SPEI bin (Figure S4) shows that the two groups have nearly identical temporal coverage with data found in almost
all years within most SPEI bins, which indicates the separation does not cause temporal inconsistency. We admit
that this separation relies on the accuracy of fire emissions and cannot rule out the effects of the long-range
transported OA from other regions, especially for the widespread drought events. As a result, it may overestimate
OA values with no local fire occurrence. With this caveat in mind, we calculated the local fire effects as the
difference between OA with and without fire emissions within each drought bin. Under the wettest conditions, there
is a minor difference of 0.23 $\mu g\,m^{-3}$ between OA with and without local fire effects, while this number becomes
four to eight times higher under droughts (SPEI < zero). The local fire-affected OA with one unit decrease of SPEI
also increases by 0.34 $\mu g\,m^{-3}$ faster than that without local fire occurrence. This illustrates the considerable
contributions of local wildfire emissions to the changes of OA under droughts. Other processes, such as long-range
transported aged OA and locally produced BSOA, may also contribute to the differences if their contributions
correlate with local fire emissions.
In summary, there is an increasing sensitivity of OA to sulfate as drought conditions worsen in the SEUS, driven by
the heightened biogenic VOC emissions and the subsequent formation of IEPOX SOA. Sulfate levels also rise under
droughts, influenced mainly by the reduced precipitation and the potentially increased aqueous and gas-phase sulfate
production. In the PNW, OA and OA wildfire emissions exhibit a close correlation, indicating that wildfire
emissions significantly drive higher OA concentrations therein.

### 3.3 CMIP6 Models Simulated Organic Aerosol Response to Drought

In this section, we evaluated the surface OA concentrations from ten CMPI6 models regarding their capability in predicting the observed SPEI-OA relationships over the CONUS during JJA 1998-2014. OA values from each model were interpolated linearly to match the spatial resolution of the gridded observational dataset. Figure 6a-j show the spatial distributions of the slopes between SPEI and OA simulated by each model. Compared with the observed slopes in Figure 2a, all models capture the strong negative slopes of more than 2 $\mu g\,m^{-3}$ per unit decrease of SPEI in the PNW region except for GFDL-ESM4 which shows a much smaller slope of less than 1 $\mu g\,m^{-3}$ per SPEI. This indicates the CMIP6 models correctly represent the sign and magnitude of the changes in OA fire emissions with droughts. By contrast, all the models have difficulties in reproducing the observed linear relationships between OA and SPEI in the SEUS. Compared to the significantly negative slope from observations, most of the models display insignificant or even positive slopes in the SEUS. BCC-ESM1, MRI-ESM2-0, and Nor-ESM2-LM show negative slopes only in part of the SEUS grids.

We also evaluated model predicted average OA enhancement under server droughts relative to non-drought periods in PNW and SEUS (Figure 6k). In the PNW region, CESM2-WACCM simulates an increase of OA mass concentration by 2.20 $\mu g\,m^{-3}$, closest to the observed value of 2.41 $\mu g\,m^{-3}$, followed by UKESM1-0-LL and CNRM-ESM2-1 with an enhancement of 1.74 $\mu g\,m^{-3}$ and 1.64 $\mu g\,m^{-3}$, respectively. GFDL-ESM4 shows the highest underestimation of the OA enhancement by 2 $\mu g\,m^{-3}$ (83%), consistent with its smallest slopes shown in Figure 6e. Smaller underestimations are found in other models, ranging from 0.96 $\mu g\,m^{-3}$ (40%) for MRI-ESM2-0 to 1.4 $\mu g\,m^{-3}$ (58%) for EC-Earth3-AerChem. In the SEUS, all the ten models underpredict the observed OA increase of 0.57 $\mu g\,m^{-3}$ with the two lowest underestimations of 0.21 $\mu g\,m^{-3}$ (37%) and 0.27 $\mu g\,m^{-3}$ (47%) found for Nor-ESM2-LM and MIRCO6, respectively. The other eight models show marginal OA enhancements between 0.02 $\mu g\,m^{-3}$ to 0.21 $\mu g\,m^{-3}$ or even a decrease (GISS-E2-1-G), indicating the incapabilities of these models in predicting OA changes in the SEUS under droughts.

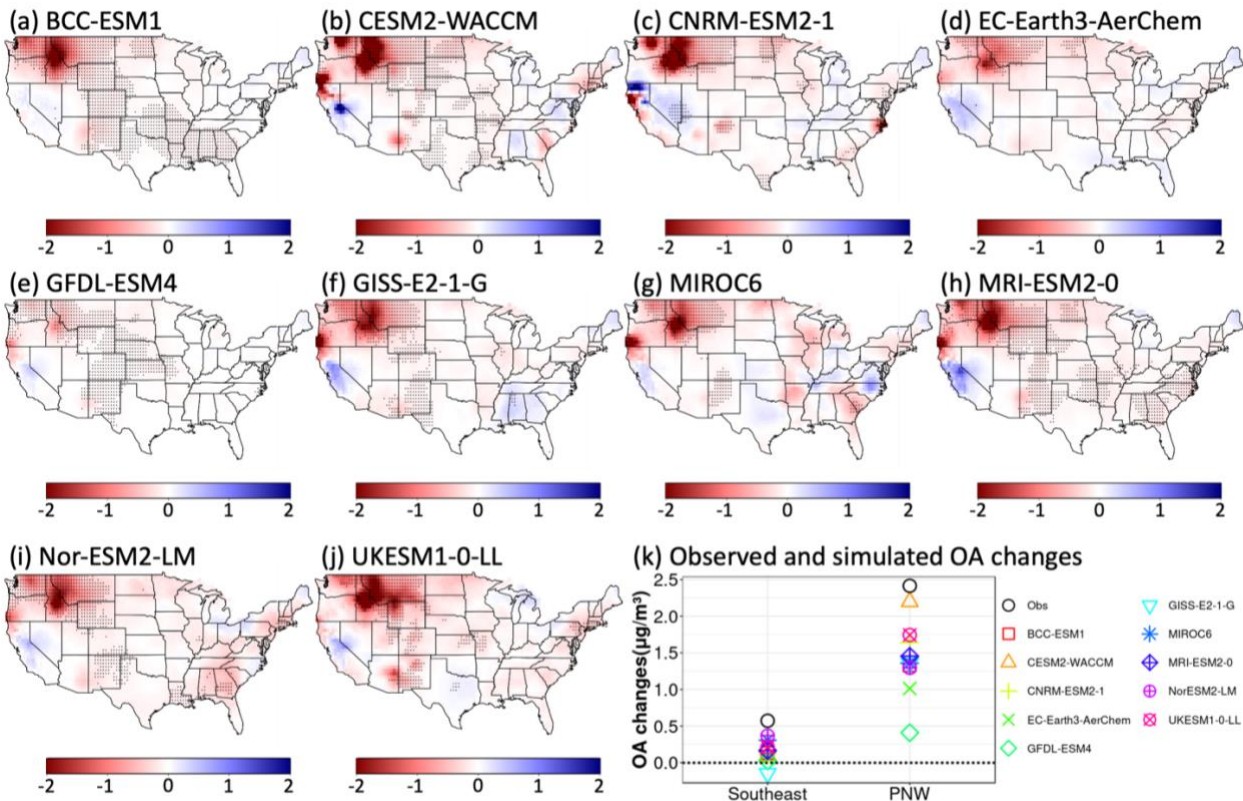

**Figure 6. (a-j) Slopes between CMIP6 model simulated OA and SPEI from 1998 to 2014 during summertime with black**
**dots indicating the P-values less than 0.05. (k) Observed and simulated OA changes under severe droughts relative to non-**
**drought conditions during the same study period in the PNW and SEUS regions.**
The poor model performance in capturing the OA changes under severe drought in the SEUS inspires us to conduct
a further regional analysis following Section 3.2. The observed and simulated changes of SEUS-mean OA, sulfate,
and their slopes within each SPEI bin are shown in Figure 7a-c, respectively. The modeled slopes are calculated in
the same way as observations (Figure 3a) and the associated scatter plot is shown in Figure S5. For the absolute OA
mass concentrations, UKESM1-0-LL has the best predictions with a less than 0.5 $\mu g\,m^{-3}$ mean bias in each SPEI
bin. CESM2-WACCM, CNRM-ESM2-1, EC-Earth3-AerChem, MICRO6, and NorESM2-LM overestimate OA
values, while the other four models show an underestimation. For the sensitivity of OA to droughts, NorESM2-LM
performs the best with an increase rate of 0.13 $\mu g\,m^{-3}$ per unit decrease of SPEI, although the rate is only 50% of the
observed value of 0.25 $\mu g\,m^{-3}$. This is consistent with the result that this model has the lowest underestimation of
OA enhancement under severe droughts. Higher underestimations of the OA sensitivity to droughts are found in
MRI-ESM2-0, BCC-ESM1, and GFDL-ESM4 with a respective change rate of 0.09 $\mu g\,m^{-3}$, 0.06 $\mu g\,m^{-3}$ and 0.02
$\mu g\,m^{-3}$ per SPEI. On the contrary, GISS-E2-1-G simulates a decrease in OA by 0.04 $\mu g\,m^{-3}$ per unit decrease of
SPEI, which is consistent with the negative OA changes under severe droughts. The rest of the models do not have a
statistically significant change rate of OA with droughts at a 95% confidence level.

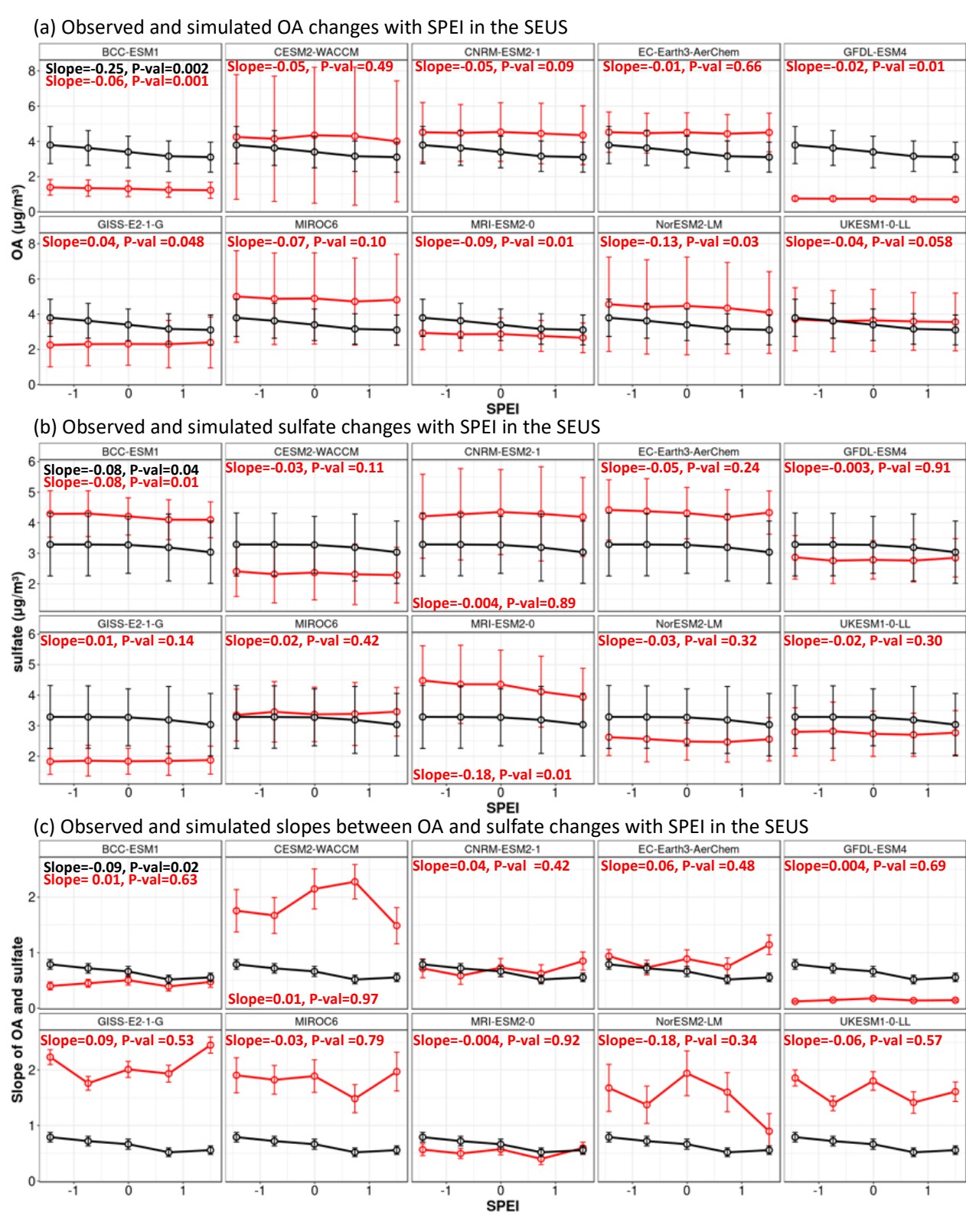

**Figure 7. SPEI bin-averaged values of OA (a), sulfate (b), and slopes of OA and sulfate (c) from observations (black lines)**
**and simulations (red lines) in the SEUS. Vertical bars indicate one standard deviation. The numbers in each subplot indicate**
**the slopes (Slope) and P-values (P-val) of the linear regression between each variable and SPEI.**


As described in Figure 3, the increase of OA under droughts in the SEUS is due to the concurrent increase of sulfate
and biogenic VOC emissions. To investigate if the models have this mechanism, we also evaluated the modeled
sensitivities of sulfate and the OA-sulfate slopes to SPEI. Only two models, BCC-ESM1 and MRI-ESM2-0, have
statistically significant increase rates of sulfate with the decrease of SPEI, despite their overestimation of ~1 $\mu g\,m^{-3}$
(30%) in terms of the absolute sulfate concentrations. BCC-ESM1 predicts the same change rate as observations
with a value of 0.08 $\mu g\,m^{-3}$ per unit change of SPEI, while MRI-ESM2-0 predicts a rate of 0.18 $\mu g\,m^{-3}$, more than
doubled the observed rate. For the slopes between OA and sulfate, however, all models cannot reproduce the
observed increase rate of 0.09 per unit decrease of SPEI. This suggests either an insensitivity of biogenic VOC
emissions in response to droughts or a lack of explicit aqueous chemistry for SOA formation in the models. For a
further investigation, we summarized how SOA is treated in each model (Table S1). In fact, SOA schemes in the 10
CMIP6 models are simplified to reduce computational cost as the climate models need to perform hundreds of years
of simulations with many ensemble members (Eyring et al., 2016). BCC-ESM1 and CESM2-WACCM use a
volatility basis set (VBS) approach that categorizes VOCs based on their volatility and simulates the chemical aging
process that leads to the formation of SOA. In CNRM-ESM2-1, SOA is prescribed from a monthly inventory
without inline calculation. EC-Earth3-AerChem, GISS-E2-1-G, and MIROC6 include the two-product scheme, in
which VOC oxidation leads to non-volatile and semi-volatile products. The rest of the models assume a fixed
percentage of yield from the emissions of VOCs. In short, the heterogeneous formation of IEPOX SOA through
reactive uptake on aqueous sulfate is not parameterized in the models. Therefore, the linear relationship between OA
and sulfate in the models is not indicative of the mechanistic dependence of OA on sulfate as demonstrated in
observations. Similar anthropogenic sources (e.g., fossil fuel combustion) and photochemical oxidants (e.g., O3 and
OH) leading to the simultaneous production of sulfate and OA can also result in positive correlations (Zhang et al.,
2011). The lack of the IEPOX SOA formation mechanism further explains why the enhancements of OA in the
SEUS are barely captured by these models. To sum up, most of the models can represent the linear relationships
between OA and SPEI in the PNW region with CESM2-WACCM and GFDL-ESM4 performing the best and worst
in predicting the OA enhancement under severe droughts. However, all the models face challenges in capturing the
OA increases under droughts in the SEUS, with Nor-ESM2-LM and MIRCO6 showing relatively better
performance indicated by their lower underestimation of OA enhancement. These challenges are mainly caused by
the lack of parameterizations of the aqueous formation of IEPOX SOA and the model deficiencies in capturing the
increase pattern of sulfate as drought intensifies.
**4 Conclusions**
In this study, the changes in organic aerosol (OA) in response to drought in the CONUS were examined. We first
displayed the spatial patterns of OA under non-drought and severe drought conditions and found most of the
CONUS experiences an abnormally higher level of OA by an average of 0.72 $\mu g\,m^{-3}$ relative to wet and normal
conditions. Regionally, the highest average increase occurs in the PNW and SEUS areas by 1.79 $\mu g\,m^{-3}$ (112 %) and
0.92 µg m$^{-3}$ (33 %), respectively. The concurrent enhancement of wildfire OA emissions in the PNW and sulfate in
the SEUS provides more insights into an in-depth investigation over these two regions.
In the SEUS, a linear regression between OA and sulfate was applied to estimate the amount of IEPOX SOA and
other OA. Results from this simplified method indicate that the IEPOX SOA drives the increase of total OA from
wet to dry conditions while other OA stays stable. Both the increase of biogenic VOC emissions and sulfate under
droughts lead to the enhancement of IEPOX SOA. Data from the NADP network shows that up to 62% lower
precipitation under droughts induces slower sulfate wet deposition rates and thus leaves more sulfate in the
atmosphere. Higher sulfate wet concentration in the precipitation indicates more in-cloud and/or gas-phase sulfate
production under droughts since cloud cover and liquid content do not show a strong sensitivity to droughts.
In the PNW, there is an overall increase of $1.44 \times 10^7$ g in the monthly OA wildfire emissions per unit decrease of
SPEI, which is the main driver of the elevated OA. There is a plateau of the OA fire emissions with SPEI between -
1.5 and -1, followed by a drop with SPEI less than -1.5. This implies that wildfire activities are not linearly related to
moisture and are also limited by the availability of fuel load. Dividing OA into groups with or without local fire
influence, we found that local fire events can increase the OA concentrations by four to eight times relative to those
without fire activities. Future work is needed to further investigate the changes in OA from other sources, such as
long-range transported OA and BSOA, in this region.
The evaluation of surface OA concentrations from ten CMIP6 models provides valuable insights into their predictive
capabilities in capturing the observed relationships between SPEI and OA over the CONUS. All the models are
found to successfully capture the negative slopes in the PNW area, indicating correct sensitivities of OA wildfire
emissions to droughts in these models. However, deficiencies are revealed in the SEUS with most models displaying
insignificant or positive slopes between OA and SPEI as opposed to significantly negative slopes from observations.
The assessment of average OA enhancement during severe droughts relative to non-drought periods further
underscores the models' varying degrees of accuracy in simulating OA response to drought. In the PNW, CESM2-
WACCM stands out with its simulated OA increase of 2.20 µg m$^{-3}$ being closest to the observed value of 2.41
µg m$^{-3}$, while GFDL-ESM4 exhibits the highest underestimation of OA enhancement by 2 µg m$^{-3}$ (83%). In the
SEUS, all models consistently underpredict the observed OA increases, highlighting their limitations in predicting
OA changes in this region under drought conditions. These limitations can be mainly attributed to the insensitivities
of sulfate to SPEI and the model deficiencies in the parameterization of the IEPOX SOA dependence on inorganic
sulfate.
This study reveals the key drivers of the enhanced OA mass concentrations in the CONUS, including higher wildfire
emissions and the simultaneous increase in biogenic VOC emissions and inorganic sulfate, which highlights the
complex physical and chemical processes involved in the aerosol composition changes under droughts. The
discrepancies in simulating OA enhancements during severe droughts underscore the need for ongoing model
improvement, particularly in accurately representing the emissions of biogenic isoprene and monoterpene, the life
cycle of sulfate, and their intricate interactions. Addressing these limitations will be crucial for enhancing the
reliability of climate models and their ability to predict the impact of future droughts on atmospheric composition
and air quality in the CONUS.

**Data availability**

Monthly SPEI data is obtained from https://spei.csic.es/spei_database_2_6 (Vicente-Serrano et al., 2010).
Observations from the IMPROVE and NADP network are downloaded from
https://views.cira.colostate.edu/fed/QueryWizard/ (FED, 2023). GFED4 wildfire emission inventory and MODIS
satellite cloud cover data are archived at https://www.geo.vu.nl/~gwerf/GFED/GFED4/ (Giglio et al., 2013) and
https://asdc.larc.nasa.gov/project/CERES/CER_SSF1deg-Month_Terra-MODIS_Edition4A (NASA, 2015),
respectively. The CMIP6 model outputs are publicly available online from the Earth System Federation Grid nodes.

**Competing interests**

The authors declare that they have no conflict of interest.

**Author contributions**

YW conceived the research idea. WL conducted the analysis. Both authors contributed to the preparation of the
manuscript.

**Acknowledgments**

The authors acknowledge researchers from the IMPROVE and NADP networks for making surface aerosol mass
and deposition observations. We thank individuals and groups from the Climatology and Climate Services
Laboratory for creating the SPEI dataset. The authors also thank the modeling groups participating in the CMIP6
AerChemMIP project for making the surface aerosol species outputs available.

**Financial support**

This research has been supported by the National Oceanic and Atmospheric Administration through the
Atmospheric Chemistry, Carbon Cycle and Climate (AC4) Program (grant no. NA19OAR4310177).

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
