# Peer review of "Observed and CMIP6 model simulated organic aerosol response"

_EGUsphere, 2024_

## Author Comment (AC1)

**Reply to Reviewers**

We sincerely appreciate all the reviewers for their constructive comments to improve the manuscript. Their comments are reproduced below followed by our responses in blue. The corresponding edits in the manuscript are highlighted with track changes.

**Reviewer #1:**

General Comments:

This study focused on the organic aerosol response to drought in the contiguous United States during the summertime. The authors have carried out a long-term analysis and recognized the different responding mechanisms in SEUS and PNW. The study also proposed a direction of model improvement through a comprehensive analysis of the ability of CMIP6 models to capture the correlation between OA and drought. The article is well-organized, but some problems still need to be addressed before consideration of publication.

Specific Comments:

(1). Line 51-52: The statement is confusing. SOA is a component of PM2.5, so why could it lead to a higher mortality rate than overall PM2.5?

**Response**: This statement is concluded from Pye et al. (2021). They demonstrated that the association between SOA and cardiorespiratory mortality was independent of total $PM_{2.5}$ mass and much stronger than that of sulfate, ammonium, nitrate, sea spray, dust, and soot. We agree that the statement may cause confusion so that we changed it to "OA is found to be associated with a higher county-level cardiorespiratory mortality rate than other major $PM_{2.5}$ components, such as sulfate, ammonium, and nitrate" in the revised manuscript.

(2). The abstract and introduction stated that the study period was 1998 to 2019, but in 2.1 and 2.2, the datasets only extend to 2018. In 3.1, line 124 and line 137 also pointed to different time periods. Please clarify the exact time period.

**Response**: Thanks for catching that. We processed the OA data up to the year of 2019, but the analysis was limited by the time span of SPEI. We changed the study period to between 1998 and 2018 across the manuscript.

(3). Line 99-101: Sites with data records longer than 5 years were selected for data analysis in this study. However, sites with data records of shorter time range need interpolation of longer time range, which may cause larger uncertainty. Could you provide the distribution of site data records according to their duration? For example, what percentage of sites have data records spanning 6 years, 7 years, etc.?

**Response**: We did not interpolate the data temporally from a shorter period to a longer period. We simply dropped the sites without data records for a certain day when conducting the spatial interpolation. As suggested, we calculated the percentage of the sites with data records greater or equal to certain years during the 21-year study period (Figure R1). There are a total of 175 sites selected for interpolation with a minimum of 5-year data record, 27% of which covers the full period. Nearly 79% of the sites have data records equal to or greater than 15 years, suggesting small uncertainties caused by the spatial interpolation. This can be further verified by the similar results between the gridded and on-site data shown in Figure 1b. Figure R1 was inserted as Figure S1 with the related texts added in Lines 105-107.

[Figure]

**Figure R1**: Percentage of sites with data records greater or equal to certain years. There are a total of 175 sites selected for interpolation with a minimum of 5-year data records during the study period.

(4). Line 124: What's the frequencies of non-drought and drought summer during the study period? Is there large interannual variation of OA concentration in either of the cases? What's the uncertainty of the mean OA concentration in these two cases?

**Response**: The maps of the frequency for non-drought and severe drought months during the study period are shown in Figure R2a. The western US states along the Rocky Mountains exhibit the highest severe drought frequency of up to 25%, while wet and normal conditions are more common in the eastern US and southern California with a frequency of more than 80%. To investigate the variation and uncertainty of OA, we calculated the standard deviation of OA under non-drought and severe drought conditions at each grid (Figure R2b). Most of the SEUS grids show low standard deviation values of ~1 $\mu g\ m^{-3}$ relative to their mean values of ~4 $\mu g\ m^{-3}$, indicating a small variation and uncertainty. By contrast, much higher standard deviations are found in the PNW region, which is expected due to the occurrence of wildfire. However, these extreme values caused by wildfire are not outliers leading to the bias of the mean, but major factors responsible for the increase of OA under droughts. Figure R2 was added to the supplemental file as Figure S2 with the related texts added in Lines 135-139.

[Figure]

**Figure R2**: (a) Maps of the frequency for non-drought (left) and severe drought (right) summer months during the study period. (b) Same as a, but for the standard deviation of OA.

(5). Section 3.2: Could you please provide a map showing which grids in the SEUS and PNW regions were included in the data analysis?

**Response**: The grids included in the SEUS (blue) and PNW (red) regional analysis are shown in Figure R3. They cover the area of 25-37°N, 75-100°W and 42-50°N, 105-125°W, respectively.

We also delimitated them in Figure 1 and 2 using boxes. The latitude and longitude boundaries were further added in Line 142-143 where we mentioned the regions for the first time.

[Figure]

**Figure R3**: The map of grids included in the PNW (red) and SEUS (blue) regional analysis.

(6). Figure 5: Wildfire OA transports over long distances. What is the necessity to distinguish between with and without local fire within a single grid? If the same grid is classified into different categories in different months and years, will there be inconsistency in calculating the average OA with/without local fire?

[Figure]

**Figure R4**: Time series of averaged OA in the PNW region separated into periods with (red) and without (blue) local fire emissions within each SPEI bin (panel). The respective trend (Slope) and P-value (P-val) of the trend are listed in each panel.

**Response**: We did such separation for PNW only, the major source region of wildfires in the US, where local fires are expected to outweigh the influence of long-range transported fire on surface OA. The intention of separating PNW into periods with and without local fire is to highlight the significant contributions from wildfire to the increase of OA relative to the conditions without wildfire occurrence, although we understood and pointed out in the main text (Line 305-307) the caveat of this method due to the transport of fire smoke. To investigate whether this separation

will cause temporal inconsistency, we examined the time series of OA grouped by periods with and without wildfire emissions within each SPEI bin (Figure R4). It shows that the two groups, represented by the blue and red lines, have nearly identical temporal coverage with data found in almost all years except for the wettest SPEI bin, which indicates the separation does not cause temporal inconsistency. In addition, OA in the PNW does not show significant trends (slopes and P-values listed in Figure S4) at a 95% confidence level in most of the SPEI bins except for the slightly increasing trend when SPEI is between -1 and 0. Therefore, OA trends will not lead to big inconsistencies in calculating the average of OA with and without local fire either. Figure R4 was added to the supplemental file as Figure S4 with the related texts inserted in Lines 302-304.

(7). This study covers a long time scale, during which anthropogenic emissions of OC, SO2 and VOCs in the United States may all have a long-term trend of change, which may affect OA concentration. When classifying the time periods based on drought or non-drought, did you consider that these human factors might differ significantly between the two periods, which might interfere with the analysis of this study?

**Response**: As shown in Figure R4, OA does not have a statistically significant decreasing trend in the PWN region, thus barely impacting the analysis therein. In the SEUS region, there is indeed a declining OA and sulfate trend as shown in the main Figure 2 and reported by other studies (e.g., Ridley et al., 2018). To verify whether the trend will significantly affect our results in the SEUS, we reproduced the key figures (Figure 1b and 3a) in Figure R5a-b using detrended OA and sulfate. The detrend is conducted by removing the 7-year moving average from the raw data in the same month of each year following Wang et al. (2017) and Li et al. (2022). OA enhancement under severe droughts is 0.78 µg m$^{-3}$ and 1.02 µg m$^{-3}$ for gridded and on-site data, respectively, which is comparable to those values derived from raw OA data (0.92 µg m$^{-3}$ and 1.11 µg m$^{-3}$) in the SEUS area. Additionally, the linear relationships of detrended OA and sulfate are also similar to those calculated from raw OA and sulfate data in terms of the gradually increasing slope from the wettest (slope=0.18) to the driest (slope=0.48) SPEI bin. These indicate that human factors do not significantly interfere with our analysis and instead natural processes play a more important role in causing the enhancement of OA. Figure R5 was added to the supplemental file as Figure S3 with the related texts inserted in Lines 148-155.

[Figure]

**Figure R5**: (a) Statistical distributions of gridded and on-site detrended OA mass concentrations under severe drought (red boxes) and non-drought (blue boxes) conditions in the PNW (left) and SEUS region (right) (b) Scatter plot of the SPEI bin-averaged detrended sulfate and OA at each grid in the SEUS with solid lines representing the linear regressions between OA and sulfate. The corresponding linear formula of each SPEI bin is listed in the bottom-right corner. The star marks in the formula indicate the regression significance at a 95% confidence level.

**Reviewer #2**

General Comments:

Li and Wang present a modeling study of the response of CMIP6 modeled organic aerosol to drought in USA. A drought response to OA is a worthwhile study since it affects our understanding of how aerosols respond to droughts and interact with climate change. However, several points need clarifications (related to deriving IEPOX-SOA based on slopes of total OA versus sulfate in measurements) and the caveats related to mechanistic representations of IEPOX-SOA in CMIP6 models need to be described so that a reader can better understand why the models could not represent how IEPOX-SOA might increase in SE USA as drought intensifies. A clearer terminology needs to be used instead of "negative slopes versus SPEI" in terms of what this means with respect to response of OA and sulfate to intensifying drought.

**Response**: The reviewer's points are well taken. The chemical mechanism of IEPOX SOA formation with the presence of sulfate is well documented (e.g., Brüggemann et al., 2020). Thus, the changes in OA can be attributed to biogenic VOC emission and sulfate, both of which are very sensitive to droughts. By taking the regression slope between sulfate and total OA, we separated the portion of OA changes sensitive to sulfate (i.e., the IEPOX SOA) from those that are not, such as direct emissions. The higher sensitivity of OA to sulfate under drought conditions may be attributed to intensified biogenic emissions of isoprene and monoterpene during droughts (Llusia et al., 2008; Wu et al., 2015). Conversely, organosulfates of anthropogenic origin are not big contributors to the enhanced sensitivity indicated by the similar increase pattern of the slopes between OA and sulfate with SPEI using detrended data (Figure R6), which can remove the effects of anthropogenic emission changes.

The SOA formation scheme in the CMIP6 models is summarized in Table R1 as suggested in comment (8) and also added in Table S1 within the supplemental file. All models do not have the aqueous IEPOX-SOA parameterization included in order to reduce computational costs for long-term simulations, which explains their poor performance in predicting OA enhancement under droughts.

The negative slopes with SPEI are a necessary specification to follow the convention of how drought indices are defined in the literature with negative indices being drought. We highlighted the 'negative slopes with SPEI' indicate an increase of OA or sulfate as drought intensifies in Line 187-188 where the negative slopes are first mentioned to avoid confusion.

Specific Comments:

(1) Line 42: "The abnormally high temperature and low humidity under droughts can enhance the volatility and oxidation of OA". Increasing temperature will increase the volatility of OA and cause OA evaporation (reducing OA) but increased photochemistry will cause oxidation and reduce volatility (increasing SOA). Are the authors referring to OA evaporation at high temperature, or do they mean enhanced photochemical aging at high temperature and low humidity that can reduce volatility and increase SOA formation? Please clarify. Also how does drought affect hygroscopicity of OA?

**Response**: Here we intended to highlight the increased SOA formation through oxidative aging under high temperatures since the evaporated OA can further form SOA via gas-phase oxidation. Therefore, we removed the 'volatility' from the sentence to avoid confusion. It is commonly considered that OA will become more hygroscopic as it is more oxidized during its evolution

although other factors, such as functional groups, carbon chain length, and aerosol liquid water content, can also have significant impacts (Jimenez et al., 2009; Kuang et al., 2021). Thus, the enhanced oxidative capability under droughts can affect the hygroscopicity of OA.

(2) Line 48: OA from wildfires is just POA or do the authors consider SOA formed due to oxidation of VOCs emitted by wildfires?
**Response**: Both POA and SOA formed from wildfire were considered here. We changed the sentence to "In addition, dry conditions can trigger large and high-intensity wildfires, emitting more POA and VOC precursors into the atmosphere."

(3) Line 149: Do the authors consider prescribed burning in SE USA? What is its change with drought compared to non-drought conditions?
**Response**: We mainly used the GFED4 inventory to investigate the OA emission changes from fire under droughts. The version of GFED4 we used includes the burned area contributions from small fires, which increases the total amount of burned area by 75% relative to its previous version and brings the prescribed burned area estimates into closer agreement with those reported by the National Interagency Fire Center (Randerson et al, 2012). Thus, the prescribed fire burning is partly, if not all, considered in the analysis. We added these details to Line 117-121 when describing the datasets used in the study.

(4) Line 188: Increased OA per unit increase in sulfate as drought "deteriorates"? Should it be drought "intensifies"?
**Response**: Thanks for the suggestion. We changed it to 'as drought severity intensifies' in the revised manuscript.

(5). Line 212-213: In addition to IEPOX-SOA other biogenic SOA types (like monoterpene SOA), and anthropogenic SOA can also change with sulfate due to the formation of organosulfates. See following references:
https://agupubs.onlinelibrary.wiley.com/doi/full/10.1029/2019JD032253
https://pubs.acs.org/doi/abs/10.1021/acs.est.9b06751
**Response**: Thanks for the information. From the provided references, organosulfates of biogenic origin usually dominate over other types in the forested areas and those of anthropogenic sources mainly occur in polluted urban areas, especially during wintertime. Biogenic emissions of monoterpene are likely to be intensified during droughts (Llusia et al., 2008; Wu et al., 2015), which can also contribute to the higher sensitivity of OA to sulfate. We further reproduced Figure 3a using detrended OA and sulfate data in Figure R6 (same as in Figure R5b for Comment #7 from the first reviewer), which can remove the effects of anthropogenic emissions. The detrend is conducted by removing the 7-year moving average from the raw data in the same month of each year following Wang et al. (2017) and Li et al. (2022). A similar pattern of the gradually increasing slope from the wettest (slope=0.18) to the driest (slope=0.48) SPEI bin was also found, which verifies the stronger dependence of OA on sulfate under droughts is mainly caused by biogenic sources. Figure R6 was added to the supplemental file as Figure S3b with the related texts inserted in Lines 222-229 in the revised manuscript.

[Figure]

OA = 0.180* × SO₄²⁻ − 0.326
OA = 0.263* × SO₄²⁻ − 0.207
OA = 0.276* × SO₄²⁻ − 0.013
OA = 0.308* × SO₄²⁻ + 0.149
OA = 0.480* × SO₄²⁻ + 0.410

SPEI bins ▭ (-3,-1) ○ (-1,-0.5) △ (-0.5,0.5) ◇ (0.5,1) ▽ (1,3)

**Figure R6**: Scatter plot of the SPEI bin-averaged detrended sulfate and OA at each grid in the SEUS with solid lines representing the linear regressions between OA and sulfate. The corresponding linear formula of each SPEI bin is listed in the bottom-right corner. The star marks in the formula indicate the regression significance at a 95% confidence level.

(6). In addition, correlation between OA and sulfate does not necessarily imply causal mechanistic relations between OA and sulfate. These caveats need to be acknowledged.
**Response**: Although correlation does not indicate causal relationships, the chemical mechanism of IEPOX SOA formation with the presence of sulfate is well documented (e.g., the two references listed above). Thus, it is natural to attribute the changes in OA to biogenic VOC emissions and sulfate, both of which are very sensitive to droughts. In addition, we pointed out that this simplified method has limitations in that OA from other processes that can lead to the simultaneous changes of sulfate and OA, such as wildfire, are miscounted as BSOA (Line 248-251). Further analysis is needed to attribute the changes of SOA to different sources more accurately. We added these caveats to Line 217-219 in the revised manuscript.

(7). Line 236-238: How does Change in pH from 4.98 to 4.87 affect IEPOX SOA formation? Can the authors quantify this with box model calculations representative of cloud chemistry? Also does pH in wet aerosols affect IEPOX-SOA in their simulations? What is the model predicted pH trend in wet aerosols in SE USA?
**Response**: The pH value of aqueous particles in the southeastern US is about 1 and lower than the pH in cloud water (Weber et al., 2016; Pye et al., 2020). Therefore, the rate of IEPOX SOA formation is slower in cloud water compared to aerosol particles (Gaston et al., 2014). However, considering the large liquid water content of clouds, which promotes dissolution, IEPOX uptake onto cloud droplets could be significant. Although we did not quantify the IEPOX SOA formation in clouds ourselves, there is an existing study comparing the IEPOX SOA formation over a range of cloud pH using the GAMMA (Gas-Aerosol Model for Mechanism Analysis) box model (Tsui et al., 2019). In their simulation, increasing pH values in cloud water while keeping the other factors constant results in a slower rate of IEPOX SOA formation. Specifically, the formation rate increases from ~0.015 μg/m³/hr at pH of 5 to ~0.03 μg/m³/hr at pH of 4 when aerosol pH is constant at 1and the increment is much higher if aerosol pH is at 3 and 4. Additionally, the formation of IEPOX SOA in cloud water under pH ⩽ 4 is comparable to or

greater than that of aerosol pH at 1, which indicates cloud water processing can be a significant source of IEPOX SOA at pH ≤ 4.

Therefore, the change of pH from 4.98 to 4.87 in our case may only slightly enhance the formation of IEPOX SOA if assuming the aerosol pH is constant at 1 for all drought conditions. However, the average pH value of ~5 only represents the regional conditions, some sites may experience more acidic rainwater during some drought months. During the study period, we found two sites in Georgia and North Carolina with pH less than 4 and their corresponding SPEI values are –0.98 and –1.39. It is noteworthy that the pH values in the NADP network have an increasing trend and the precipitation becomes less acidic due to the reduction of sulfur emissions during the last two decades (Change et al., 2022). Extending the period to 1980, there are 17 sites found with pH values smaller than 4 during summer months and their average SPEI is –1.05. Therefore, droughts are likely to reduce the cloud pH values lower enough at some locations, thus favorable for SOA formation.

Unfortunately, we did not find the outputs of pH value or concentrations of hydrogen ion from the 10 CMIP6 models and thus we could not quantify their changes with droughts at this moment. However, these models do not consider the pH impact on the formation of IEPOX SOA because the heterogeneous chemistry is not included in their aerosol mechanisms as described below in comment (8). Thus, even if the changes of pH values with drought were simulated correctly in the models, it would not affect the modelled formation of SOA. The discussion of cloud pH impacts on IEPOX SOA formation was added in Line 276-285 in the revised manuscript.

(8). Line 270 and section 3.3: The authors should present brief discussions of how IEPOX-SOA is simulated within each of the 10 CMIP6 models. For example, do these models consider limitations of mass transfer to IEPOX gas on aqueous sulfate aerosols due to gas-phase diffusion, interfacial accommodation at gas-particle interface, diffusion limitations within SOA coatings and aqueous phase reaction rates of IEPOX forming 2-methyltetrols and organosulfates (key components of IEPOX-SOA)? Which of these models consider the effects of particle-phase (solid, liquid) as a function of relative humidity on IEPOX-SOA formation? For example, please see:
https://agupubs.onlinelibrary.wiley.com/doi/full/10.1002/2016RG000540
https://pubs.acs.org/doi/abs/10.1021/acsearthspacechem.1c00356
**Response**: This is a good suggestion. Based on the references listed in Table S1, the SOA formation schemes in the 10 CMIP6 models are simplified to reduce computational cost as the climate models need to perform hundreds of years of simulations with many ensemble members (Eyring et al., 2016). BCC-ESM1 and CESM2-WACCM use a volatility basis set (VBS) approach that categorizes VOCs based on their volatility and simulates the chemical aging process that leads to the formation of SOA. SOA in CNRM-ESM2-1 is prescribed from a monthly inventory without inline calculation. EC-Earth3-AerChem, GISS-E2-1-G, and MIROC6 include the two-product scheme, in which VOC oxidation leads to non-volatile and semi-volatile products. The rest of the models assume a fixed percentage of yield from the emissions of VOCs. In short, the heterogeneous formation of IEPOX SOA through reactive uptake on aqueous sulfate is not parameterized in the models. Therefore, the linear relationship between OA and sulfate in the models is not indicative of the mechanistic dependence of OA on sulfate as demonstrated in

observations. Similar anthropogenic sources (e.g., fossil fuel combustion) and photochemical oxidants (e.g., $O_3$ and OH) leading to the simultaneous production of sulfate and OA can also result in positive correlations (Zhang et al., 2011). The lack of the IEPOX SOA formation mechanism further explains why the enhancements of OA in the SEUS are barely captured by these models. We inserted one column in Table S1 (also reproduced below as Table R1) to describe the SOA formation in each model and rewrote the texts in Line 373-396 when describing Figure 7c.

**Table R1.** Information of the ten CMIP6 models selected for evaluation.

| Models | Resolution (lat×lon) | Aerosol scheme | SOA formation | Model references |
|---|---|---|---|---|
| BCC-ESM1 | 2.81°×2.81° | BCC-AGCM3-Chem | volatility basis set (VBS) scheme | Wu et al. (2020) |
| CESM2-WACCM | 0.95°×1.25° | MAM4 | volatility basis set (VBS) scheme | Danabasoglu et al. (2020) |
| CNRM-ESM2-1 | 1.40°×1.40° | TACTIC | prescribed from a monthly inventory | Séférian et al. (2019) |
| EC-Earth3-AerChem | 2.00°×3.00° | TM5 | two-product scheme | van Noije et al. (2021) |
| GFDL-ESM4 | 1.00°×1.25° | Bulk aerosol scheme | 10% per-carbon yield from VOCs | Dunne et al. (2020) |
| GISS-E2-1-G | 2.00°×2.50° | TCADI | two-product scheme | Kelley et al. (2020) |
| MIROC6 | 1.40°×1.40° | SPRINTARS | two-product scheme | Tatebe et al. (2019) |
| MRI-ESM2-0 | 1.87°×1.87° | MASINGAR | 14% of monoterpene and 1.68 % of isoprene emissions are assumed to form SOA | Yukimoto et al. (2019) |
| NorESM2-LM | 1.87°×2.50° | OsloAero6 | 15% and 5% yield from oxidation of monoterpenes and isoprene | Seland et al. (2020) |
| UKESM1-0-LL | 1.25°×1.87° | UKCAl | 26% yield from gas-phase oxidation of VOCs | Senior et al. (2020) |

(9). Line 275: When referring to strong negative slopes of OA per unit decrease of SPEI, the authors should clarify that this represents increase of OA as droughts intensify.
**Response**: We added some texts to explain this in Line 187-188 where the negative slope between SPEI and OA is first mentioned.

(10). Line 318: Define OA-sulfate slopes to SPEI. How is this calculated from the models? Is it slope of OA versus sulfate (let's say its X) and then the slope of X versus SPEI?
**Response**: The OA-sulfate slopes to SPEI are calculated in the same way as observations shown in Figure 3a. We first calculated the SPEI bin-averaged value of OA and sulfate at each grid cell and then conducted the linear regression spatially across all grids for each SPEI bin. To better

demonstrate this, we reproduced Figure 3a for each model in Figure R7, which was also added in Figure S5 in the supplemental file with related texts inserted in Line 350-351. The regression slopes were further extracted to plot Figure 7c.

[Figure]

**Figure R7**: Scatter plot of the SPEI bin-averaged sulfate and OA simulations at each grid in the SEUS with solid lines representing the linear regressions between OA and sulfate.

(11). Can the authors evaluate sensitivity of IEPOX SOA in models to sulfate with long term observations, for example the IMPROVE network and SEARCH sites? For example, Zheng et al. 2020: https://acp.copernicus.org/articles/20/13091/2020/acp-20-13091-2020.html
**Response**: We evaluated the modeled sensitivities of IEPOX SOA to sulfate in Figure 7c using the long term IMPROVE data as explained in the last comment (10). In Zheng et al. (2020), they averaged OA and sulfate across all the sites in the SEUS and calculated the linear regression temporally. We adopted a spatial calculation of the linear slopes for two reasons: (1) Average across all the sites/grids will significantly reduce the number of data points after the allocation among SPEI bins; (2) The regional mean of SPEI may average out some drought signals because drought is grid specific and can differ spatially within the SEUS (Ford et al., 2014). Despite the different methods used, the linear slope in our calculation (0.56) under non-drought conditions is similar to that of Zheng et al using SEARCH sites (0.51). Therefore, our linear slope calculation method reproduces the sensitivity of OA to sulfate reported by the existing studies. We added such discussions in Line 206-214 in the revised manuscript.

(12). It was shown Zheng et al. 2020 (listed above) that a global model overestimates OA sensitivity to sulfate by at least a factor of 2. But it seems the CMIP6 models are underestimating the sensitivity of OA to sulfate? How does this sensitivity change with drought? The authors should consider plotting the long-term multi-year model predicted concentrations of IEPOX-SOA versus sulfate at least in the models that explicitly model IEPOX-SOA formation as function of RH, particle acidity, sulfate, organic coatings etc. Then the drought years can be marked on this plot. The same can be done for measurements.

**Response**: We meant to compare the sensitivity of OA and sulfate between observations (black lines) and simulations (red lines) in Figure 7c as described in comment (10) and (11). If the simulated slopes (red lines) are higher than those of observations (black lines), such as CESM2-WACCM and GISS-E2-1-G, the models are overestimating the sensitivity and vice versa. The sensitivity changes with drought can be evaluated by comparing the slope changes with SPEI, which is indicated by the red (model) and black (observation) numbers in each subplot. The observed higher sensitivity of OA to sulfate is not captured by all the models. It is understandable since these models do not have the IEPOX SOA formation scheme included in their aerosol parameterization as mentioned in comment (8).

**References:**

Brüggemann, M., Xu, R., Tilgner, A., Kwong, K. C., Mutzel, A., Poon, H. Y., ... & Herrmann, H. (2020). Organosulfates in ambient aerosol: state of knowledge and future research directions on formation, abundance, fate, and importance. Environmental Science & Technology, 54(7), 3767-3782.

Chang, C. T., Yang, C. J., Huang, K. H., Huang, J. C., & Lin, T. C. (2022). Changes of precipitation acidity related to sulfur and nitrogen deposition in forests across three continents in north hemisphere over last two decades. Science of the Total Environment, 806, 150552.

Eyring, V., Bony, S., Meehl, G. A., Senior, C. A., Stevens, B., Stouffer, R. J., & Taylor, K. E. (2016). Overview of the Coupled Model Intercomparison Project Phase 6 (CMIP6) experimental design and organization. Geoscientific Model Development, 9(5), 1937-1958.

Ford, T., & Labosier, C. F. (2014). Spatial patterns of drought persistence in the Southeastern United States. International Journal of Climatology, 34(7), 2229-2240.

Gaston, C. J., Riedel, T. P., Zhang, Z., Gold, A., Surratt, J. D., & Thornton, J. A. (2014). Reactive uptake of an isoprene-derived epoxydiol to submicron aerosol particles. Environmental science & technology, 48(19), 11178-11186.

Jimenez, J. L., Canagaratna, M. R., Donahue, N. M., Prevot, A. S. H., Zhang, Q., Kroll, J. H., ... & Worsnop, D. R. (2009). Evolution of organic aerosols in the atmosphere. science, 326(5959), 1525-1529.

Kuang, Y., Huang, S., Xue, B., Luo, B., Song, Q., Chen, W., ... & Shao, M. (2021). Contrasting effects of secondary organic aerosol formations on organic aerosol hygroscopicity. Atmospheric Chemistry and Physics, 21(13), 10375-10391.

Li, W., Wang, Y., Flynn, J., Griffin, R. J., Guo, F., & Schnell, J. L. (2022). Spatial variation of surface O3 responses to drought over the contiguous United States during summertime: Role of precursor emissions and ozone chemistry. Journal of Geophysical Research: Atmospheres, 127(1), e2021JD035607.

Llusia, J., Penuelas, J., Alessio, G. A., & Estiarte, M. (2008). Contrasting species-specific, compound-specific, seasonal, and interannual responses of foliar isoprenoid emissions to experimental drought in a Mediterranean shrubland. International Journal of Plant Sciences, 169(5), 637-645.

Pye, H. O., Nenes, A., Alexander, B., Ault, A. P., Barth, M. C., Clegg, S. L., ... & Zuend, A. (2020). The acidity of atmospheric particles and clouds. Atmospheric chemistry and physics, 20(8), 4809-4888.

Randerson, J. T., Chen, Y., Van Der Werf, G. R., Rogers, B. M., & Morton, D. C. (2012). Global burned area and biomass burning emissions from small fires. Journal of Geophysical Research: Biogeosciences, 117(G4).

Ridley, D. A., Heald, C. L., Ridley, K. J., & Kroll, J. H. (2018). Causes and consequences of decreasing atmospheric organic aerosol in the United States. Proceedings of the National Academy of Sciences, 115(2), 290-295.

Tsui, W. G., Woo, J. L., & McNeill, V. F. (2019). Impact of aerosol-cloud cycling on aqueous secondary organic aerosol formation. Atmosphere, 10(11), 666.

Wang, Y., Xie, Y., Dong, W., Ming, Y., Wang, J., & Shen, L. (2017). Adverse effects of increasing drought on air quality via natural processes. Atmospheric Chemistry and Physics, 17(20), 12827-12843.

Weber, R. J., Guo, H., Russell, A. G., & Nenes, A. (2016). High aerosol acidity despite declining atmospheric sulfate concentrations over the past 15 years. Nature Geoscience, 9(4), 282-285.

Wu, C., Pullinen, I., Andres, S., Carriero, G., Fares, S., Goldbach, H., ... & Mentel, T. F. (2015). Impacts of soil moisture on de novo monoterpene emissions from European beech, Holm oak, Scots pine, and Norway spruce. Biogeosciences, 12(1), 177-191.

Zhang, Q., Jimenez, J. L., Canagaratna, M. R., Ulbrich, I. M., Ng, N. L., Worsnop, D. R., & Sun, Y. (2011). Understanding atmospheric organic aerosols via factor analysis of aerosol mass spectrometry: a review. Analytical and bioanalytical chemistry, 401, 3045-3067.

---

## Author Response (AR2)

**Reply to Reviewers**

We sincerely appreciate all the reviewers for their constructive comments to improve the revised manuscript. The technical correction suggested by the second reviewer is reproduced below followed by our responses in blue. The corresponding edits were also added to the latest manuscript.

**Reviewer #2:**

General Comments:

The authors have reasonably addressed my comments. They acknowledged that none of the CMIP6 models include the mechanistic scheme of IEPOX SOA formation, and that the correlation they see between IEPOX SOA and sulfate in the models could be due to simultaneous production of sulfate and OA (from fossil emissions and/or photochemistry).

The resulting implications is that the trend between OA and sulfate in SE USA seems to be driven by confounding factors that cause them to be correlated at least in the models used and not through the known mechanistic relations between IEPOX SOA and sulfate. Using the trend between OA and sulfate (even after detrending) is not convincing enough to call it IEPOX-SOA. This caveat needs to be clearly acknowledged in the Conclusions.

**Response**: We are aware of the limitation of the methodology used in the manuscript to derive IEPOX-SOA based on the linear relationship between OA and sulfate. We pointed it out in the manuscript in Line 247-250 "It is noted that the approximation of IEPOX SOA here is the upper limit of BSOA since other processes that can lead to the simultaneous changes of sulfate and OA, such as wildfire, are miscounted as BSOA in the calculation. Further analysis is needed to attribute the changes of SOA to different sources more accurately". In the model evaluation, we intend to highlight the model deficiency of lacking the aqueous SOA formation, which explains their poor performance in predicting OA enhancements under droughts. As suggested, we further added the caveat of the method to the conclusion section in Line 403-404.